# Estimation Based on Chirp Modulation for Desired and Interference Power and Channel Occupancy Ratio in LoRa

**DOI:** 10.3390/s22114140

**Published:** 2022-05-30

**Authors:** Osamu Takyu, Gaku Kobayashi, Koichi Adachi, Mai Ohta, Takeo Fujii

**Affiliations:** 1Department Electrical & Computer Engineering, Shinshu University, Nagano 380-8553, Japan; 20w2038e@shinshu-u.ac.jp; 2Advanced Wireless Communication Research Center, The University of Electro-Communications, Tokyo 182-8585, Japan; adachi@awcc.uec.ac.jp (K.A.); fujii@awcc.uec.ac.jp (T.F.); 3Department Electronics Engineering and Computer Science, Fukuoka University, Fukuoka 814-0180, Japan; maiohta@fukuoka-u.ac.jp

**Keywords:** LoRa, chirp modulation, LPWAN, co-channel interference

## Abstract

In terms of low power consumption and long-range communication—low-power wide-area networks (LPWAN) are suitable for wireless sensor networks. Long-range (LoRa) wireless communication is one of the standards of LPWAN. LoRa shares common frequency spectrum bands with both multiple transmitters, which are the sensors in the LoRa system (and those in the other system). Therefore, co-channel interference (CCI) degrades the packet delivery rate. To avoid CCI, the CCI power and the occurrence probability of CCI in the target channel are estimated, then the sensor decides whether to use the channel and where the occurrence probability of CCI is defined as the channel occupancy ratio (COR). If a large signal power is obtained at the receiver, the received signal can be demodulated because of the capture effect. The desired signal power must also be estimated for the capture effect. In this study, we propose an estimation scheme based on chirp modulation of LoRa under spectrum sharing among other systems. The proposed scheme estimates the desired signal power, CCI power, and COR. From the computer simulation results, we clarify the advantages of the proposed scheme in terms of estimation accuracy and packet delivery rate.

## 1. Introduction

Low-power wide-area networks (LPWAN) achieve long-range communication with low power and are one of the most powerful wireless sensor networks for the Internet of Things [1]. Long-range (LoRa) wireless communication, which is an LPWAN, employs a distributed wireless access scheme and can be utilized to construct a wireless communication link in various locations, such as wireless local area networks [2]. The wireless sensor networks constructed by the cellular system, such as massive machine-type communications for 5G [3] and narrowband IoT [4], are also attracting attention. However, their high costs and the requirements of wireless licenses are the main hindrances to their deployment. On the other hand, the LoRa has an advantage in terms of easy deployment due to the use of unlicensed bands and low costs.

If the distance between different LoRa systems is short, the co-channel interference (CCI) between systems degrades the packet delivery rate (PDR) [5]. In Japan, the LoRa is assigned to the 920 MHz frequency band. It is a specified low-power radio station and shares common frequency bands with smart utility network(s) (SUN) [6]. Therefore, LoRa also obtains the CCI from other systems [7]. For a high PDR performance, analyses of the wireless communication link quality and wireless access scheme for suppressing the CCI are required [5].

To analyze the wireless communication link quality, the occurrence frequency of CCI, power of CCI, and power of the desired signal power should be specified. The occurrence frequency of CCI is useful for exploiting a channel with zero or few CCIs [8], where it is referred to as the channel occupancy ratio (COR) [9]. Channel selection based on the estimated COR improves the throughput performance [10]. If the powers of the CCI and desired signals are estimated, successful demodulation—even in the presence of CCI—is expected, which is referred to as the capture effect [11]. Exploiting the channel with no CCI and demodulating the packet owing to the capture effect could realize frequency sharing among the other systems and improve the usage efficiency of the frequency spectrum. To estimate the power of the desired signal, the construction of a communication link is necessary; however, the accuracy of estimating the power of the desired signal degrades owing to the CCI. Therefore, an estimation scheme (for the power of CCI, the power of the desired signal, and the COR with robustness to the CCI) is essential.

LoRa employs chirp modulation, which is a spread spectrum technique, to ensure robustness to the CCI [12]. Chirp modulation alters the frequency sweep pattern of the carrier wave (depending on the digital data). If the frequency components of the transmitted signal are detected by a period shorter than the sweeping period of the carrier wave, they are separated into two components: the frequency components occupied and unoccupied by the carrier wave, where the former and latter are referred to as the partially occupied bands (POD) and partially unoccupied bands (PUD), respectively. POD includes the components of the desired signal, CCI, and noise, and PUD includes that of the CCI and noise. If the residual components are ignored, POD minus PUD causes the components of the desired signal; thus, the power of the desired signal can be estimated. Because the residual components have a tendency similar to that of noise, they can be compressed by the averaging process. The components of the POD are switched between the component of the noise and that of the noise plus the CCI by the occurrence of the CCI, where the CCI occurrence is decided by the access of the other system sharing common frequency bands. The average occurrence probability of CCI is determined by the COR of the other system [13].

Packet access from each sensor to the data center is assumed, where the packet comprises multiple modulated symbols. When the components of POD are estimated from various packets with different arrival times, the estimated component sets are the mixed data between the noise components and noise plus CCI components. Therefore, the statistical analysis can separate the two probability distributions of the noise and noise plus CCI and can estimate the mixing rate between them, where the mixing rate is equivalent to the COR.

This study proposes an estimation scheme for the desired signal power, CCI power, and COR based on the chirp modulation scheme of LoRa. In the proposed scheme, the chirp pattern selected by the transmitter is estimated from the tentative chirp demodulation. In accordance with the estimated chirp pattern, the POD and PUD are estimated from the spectrum of the received signal. The desired signal components are specified by the differentials from the POD to PUD, and the desired signal power is then estimated from them. Multiple PUDs are estimated from multiple packets; thus, their dataset is composed of the mixed data between the noise components and noise plus CCI components. Based on the assumption that the dataset is modeled by the random process of the double Gaussian mixture probability density functions, we estimated each Gaussian probability density function and mixed rate using an expectation–maximization algorithm (EM algorithm) [14]. From the estimation result, the CCI power, noise power, and COR were estimated. In the proposed scheme, the estimation error of the chirp pattern degraded the estimation accuracy. A cyclic redundancy check (CRC) was utilized to determine the demodulation accuracy of the packet. If the CRC detects a packet error, the PUD estimated from the packet is deleted; however, the measured PUD does not involve random sampling. In particular, as the CCI increases, packet errors occur more frequently. Therefore, more estimated samples with larger CCI power are deleted; thus, the accuracy of estimating CCI power is degraded. In the proposed scheme, if the CRC detects a packet error and the tentatively estimated CCI power exceeds a certain threshold, the measured PUD is still employed for the estimation. Because the occurrence of the packet error is caused by a large CCI, we assume that the estimation result for a larger CCI power is more likely to be accurate. We employed the tentative estimation results of the mean and deviation of the CCI power estimated from all the packets with no error decided by the CRC to determine the threshold.

The specific contributions of this study are as follows:An estimation scheme of the desired signal power, CCI power, and COR based on the chirp modulation of LoRa is proposed.A packet selection based on the estimated CCI power is proposed to improve the estimation accuracy.A channel selection criterion for maximizing the PDR performance is constructed.

We showed the proposed estimation scheme of the desired signal power, CCI power, and COR based on the chirp modulation of LoRa [15,16,17]. This paper explains the mechanism of the proposed estimation scheme in detail and adds the extensive simulation performance evaluations for clarifying the advantage of the proposed estimation scheme. However, an access scheme based on the estimated power and COR is not indicated. This study explains the detailed mechanism of the proposed scheme, indicates the access protocol based on the estimated power and COR, and evaluates the PDR of this access scheme to clarify the advantages of the proposed scheme.

An overview of this study is as follows: Section 2 presents the conventional studies of our considered estimation scheme. We describe the system model assumed in Section 3. Section 4 explains the proposed estimation in detail. Section 5 presents the channel-selection scheme based on the estimated power and COR. The numerical results obtained by the computer simulation are presented in Section 6. Finally, Section 7 concludes the paper.

## 2. Related Works

The estimation schemes of the signal power, CCI power, and COR were considered [18,19,20,21,22,23,24,25,26,27,28,29,30,31,32,33,34,35,36,37,38,39,40,41]. In the spectrum sharing of LPWA, the COR of the other system is estimated by comparing the occupied time period of the other system with a certain time threshold [18]. To determine the threshold, the interference power and noise power should be estimated; however, these are not considered. The estimation of COR based on a support vector regression under the CCI with various powers is proposed [19]. Its accuracy is low under the low CCI power. During the accessing of the signal, the CCI power is smaller than the signal power and, thus, the high accuracy of estimation is not achieved. The compensation scheme of estimating COR to the spectrum sensing error is proposed [20]. The average false alarm and miss-detection probability are required (but how to estimate them is not considered).

The pilot signal-assisted estimation of the signal power was considered [21,22,23]. The estimation of CCI power under electromagnetic compatibility [21], pilot signal construction for estimating the desired to undesired power ratio under the down link [22], and estimation of the CCI power to the primary receiver using the symmetrical feature of the time division duplex [23], were proposed. Time synchronization is required for detecting the pilot signal, and the frame format of the CCI signal should be specified. Therefore, an estimation scheme for various CCIs is not available. The detection scheme based on the stochastic resonance for the weak LoRa signal under the LoRa and WiFi coexistence in 2.4 GHz bands has been proposed [24]. However, the effect of detecting the weak signal by the stochastic resonance is limited in the linear receiver [25]. The estimation of the CCI based on probabilistic modeling has been considered [26,27], but information on the CCI signal format is necessary. In addition, if the mathematical formula for the CCI is not determined, the probabilistic model is not utilized. An estimation scheme for CCI power based on the Gaussian model of the CCI has been proposed [28]. It estimates the probability distribution of CCI under the noise level in accordance with the Gaussian model. It is necessary to specify the occurrence of the CCI. If the occurrence of the CCI is caused by random access from the other system, the estimation error becomes large because of error recognition of the noise power, which is considered the CCI power. The noise power estimation for long-term and wideband spectrum measurements has been proposed [29]. Since (in spectrum sharing among the other systems) the noise level is fluctuated due to the interference occurring by the other systems, the long-term measurements are not suitable. The estimation of propagation loss for measuring the CCI is proposed [30]. However, the vacant time of the CCI is not considered and, thus, its exploitation by the other system is difficult.

Estimations of CCI and the desired power based on stochastic reasoning have been considered, such as the Bayesian scheme [31] and Kalman filtering [32]. In the Bayesian scheme [31], the accuracy of the estimation is at most 90%. To employ Kalman filtering, the time correlation of the CCI is required. When CCI occurs randomly, the tracking accuracy is degraded. The estimation of the cumulative CCI among several interference sources has been considered [33]. It is based on geographical information. It does not assume the occurrence probability of each interference source but a 100% occurrence. Therefore, the estimated CCI power is larger than the actual value.

To reduce the impact of CCI, successive interference cancellation [34], interference rejection combining [35], and adaptive array antennas [36] have been proposed. The accuracy of estimating the CCI and desired signal powers is improved owing to the mitigation of the CCI. The large computational complexity and body size of the estimator are challenges. The estimation of the CCI and desired power have been proposed (adopting the different bandwidth of the CCI and desired signals). When the desired and the CCI signals are the broadband OFDM and narrowband signals, respectively, the estimation of the narrowband signal power based on the Welch periodogram is proposed [37]. The estimation of the desired and undesired signal powers based on the pilot signal has been proposed [38]. Because the narrowband signal has the weakness of multipath fading owing to the lack of frequency diversity, the accuracy of the narrowband signal power is degraded. When the desired and CCI signals are the continuous wave and the frequency modulation (FM) signals, respectively, the estimation scheme of the continuous wave and FM signal power is proposed [39]; however, the estimation of the modulated signal is unavailable.

The estimation scheme of the collision-free period, which is equal to the unoccupied time period by the other system, under the frequency sharing between the Bluetooth low energy and the IEEE 802.15.4 is proposed [40]. The estimation depends on the computer simulation and, thus, performing the computer simulation; changing the situation involves large computational complexity. The spectrum sensor for TV white space is considered [41]. It achieves low computational complexity owing to the simple detection of CCI but cannot estimate the power of CCI.

To the best of our knowledge, the estimation scheme of the desired power, CCI power, and COR with high accuracy under spectrum sharing among other systems has not yet been considered.

## 3. System Model

Figure 1 illustrates the assumed wireless sensor networks. A long-range wide-area network (LoRa), which is a low-power wide-area network, is assumed. There is a data center and sensors within the cover area constructed by the data center. Each sensor accesses the data center to inform the sensing data.

We assumed that each sensor transmitted data within a certain time period. Figure 2 shows the wireless accessing image of each sensor. Periodic data transmission is suitable for detecting changes in the monitoring environment [42]. In the wireless access scheme, each sensor was assigned an orthogonal time period and/or a frequency-orthogonal channel [43]. In accordance with the specifications of LoRa [44], the access time limitation, which is the duty cycle, is under the 1% and, thus, packet collisions among sensors within the LoRa system rarely occur. Under this assumption, the wireless access of each sensor does not affect that of the other sensors. For simplicity, we assume that the number of sensors is one; it is noted that the extension of the proposed scheme to multiple sensors is straightforward.

We assume frequency spectrum sharing between the single-carrier (SC) and LoRa systems, where the example of an SC system is a wireless smart utility network (Wi-SUN) [6]. Various channels of Wi-SUN are defined. We assume that the channel bandwidth of the SC is approximately equal to or a few tens of percentage points smaller than that of LoRa. We also assume that if packet collision between SC and LoRa occurs, the entire packet of LoRa overlaps with the packet of the SC system. If a partial overlap of packets between LoRa and SC occurs, the estimated signal power of CCI is reduced. We assume the fluctuation of the power of CCI by the shadowing effect. In the performance evaluation, the impact of partial overlap is also considered.

Figure 2 shows the image of spectrum sharing between the sensor and SC system in the multiple channel environment. In each channel, the time period of the channel occupied by the SC system is random. In addition, the power of CCI emitted by the SC system depends on the distance between the SC system and the data center, and then it fluctuates in each channel. If the SC system and the sensor simultaneously access the common channel and the power of the CCI is larger than a certain level, the packet collision occurs and, thus, the demodulation of the data from the sensor is failure. If it is not, as in channel 2 of Figure 2, the demodulation of the data from the sensor is successful owing to the capture effect [11]. If each sensor exploits the channel with low COR as well as the low power of CCI, the packet delivery rate (PDR) becomes large. For selecting the suitable channel for each sensor, this paper considers the estimation of the desired signal power, the CCI power, and the COR.

The transmitted signal is modulated using a chirp modulation scheme. In the chirp modulation scheme, the carrier frequency cyclically increases or decreases, and the change in carrier frequency is determined by the data bits. The altering pattern of the carrier frequency is defined as the chirp pattern. The number of chirp patterns, *L*, is $2^{N_{SF}}$ [12], where $N_{SF}$ is the spreading factor, which is a parameter for spreading the carrier signal over the frequency domain within one transmitted symbol time. The number of transmitted bits *B* is
(1)$$\begin{array}{c} B=\log_{2}L=\log_{2}2^{N_{SF}}=N_{SF}. \end{array}$$

As $N_{SF}$ increases, the number of bits per modulated symbol also increases. When we define *W* and T=1/W as the occupied bandwidth and sampling periods, respectively, the time length of one modulated symbol, $T_{s}$, is $2^{N_{SF}}T$ [12]

Figure 3 illustrates an image of chirp-modulated symbols. In this figure, we define the discrete values of the time and frequency axes, where the minimum resolution of the frequency is $W/(2^{N_{SF}})$, and the frequency bins are assigned to the label as $k=1,2,\ldots ,2^{N_{SF}}$. In addition, the minimum time resolution is $T_{s}/2^{N_{SF}}=T$, and each time span is assigned a label of $n=1,2,\ldots 2^{N_{SF}}$. In the next time span after $n=2^{N_{SF}}$, the label returns to n=1; therefore, the label of the time span is periodic. We define $X_{n,k}$ as the partial frequency spectrum of the *k*th channel and *n*th time span.

We also define the group of all channels and time span as ***V***, which is defined as follows: (2)$$\begin{array}{c} V=\left\{\left(1,1\right),\left(1,2\right),\ldots ,\left(n,k\right),\ldots ,\left(2^{N_{SF}},2^{N_{SF}}\right)\right\}. \end{array}$$

The chirp pattern is modeled as a combination of partial frequency spectra. We define $\phi_{l}$ as the spectral combination of the *l*th chirp pattern, where $l=1,2,\ldots ,\;L=2^{N_{SF}}$. In the first modulated symbol of Figure 3, the chirp pattern is l=2 and its spectrum combination is
(3)$$\begin{array}{c} \phi_{l=2}=\left\{(1,2),(2,3),(3,4),(4,1)\right\}. \end{array}$$

In the demodulation, the power of the partial spectrum, $|X_{n,k}{|}^{2}$, in accordance with the combination of the chirp pattern, $\phi_{l}$, is combined. The combined results for the partial frequency spectrum in $\phi_{l}$ are given as
(4)$$\begin{array}{c} \eta_{l}=\sum_{\left(n,k\right)\in \phi_{l}}{\left|X_{n,k}\right|}^{2}. \end{array}$$

In the first modulated symbol of Figure 3, the combined results for the partial frequency spectrum are
(5)$$\begin{array}{c} \eta_{l=2}=|X_{1,2}{|}^{2}+|X_{2,3}{|}^{2}+|X_{3,4}{|}^{2}+{\left|X_{4,1}\right|}^{2}. \end{array}$$

The receiver estimates the chirp pattern number selected by the transmitter $l^{☆}$ as
(6)$$\begin{array}{c} l^{☆}=\arg_{l}\max\eta_{l}, \end{array}$$

In accordance with the relationship between the chirp pattern number *l* and the information data, the receiver specifies the transmitted data.

We assume a packet access scheme in which the payload of a packet comprises multiple data bits and a cyclic redundancy check (CRC). The CRC is effective for detecting error bits. In the assumed wireless sensor networks, if the CRC detects any error bits from the packet, the packet is deleted. The packet is not re-transmitted to compensate for the deleted packet. Even if CCI occurs, the received signal to noise plus interference power ratio (SINR) is greater than the required level for demodulation, and the demodulation is successful owing to the capture effect [11].

## 4. Proposed Estimation Scheme

### 4.1. Estimation of Desired Power, CCI Power, and COR

Figure 4 illustrates the process flow of the proposed estimation scheme.

#### 4.1.1. Signal Recovery

Chirp demodulation is performed for one packet. Accordingly, the chirp pattern selected by the transmitter is estimated and denoted as $\hat{\phi_{l}}$. The $N_{pkt}$ chirp patterns are estimated when the number of chirp modulation symbols per packet is defined as $N_{pkt}$. When the frequency spectrum of the received signal is obtained using the fast Fourier transform (FFT), the discrete partial frequency spectrum, $X_{n,k}$, is obtained. In accordance with $\hat{\phi_{l}}$, the two types of estimated powers, $\eta_{l,A},\;\eta_{l,B}$, are obtained as follows: (7)$$\begin{array}{c} \eta_{l,A}=\sum_{\left(n,k\right)\in \hat{\phi_{l}}}{\left|X_{n,k}\right|}^{2}/N_{\eta_{A}}, \end{array}$$(8)$$\begin{array}{c} \eta_{l,B}=\sum_{\left(n,k\right)\in \left\{V\backslash \hat{\phi_{l}}\right\}}{\left|X_{n,k}\right|}^{2}/N_{\eta_{B}}, \end{array}$$
where $\eta_{l,A}$ and $\eta_{l,B}$ are the combined results of the power whose spectrum includes—and does not include—the components of the desired signal in the *l*th chirp pattern, respectively. The partial frequency spectra of the former and latter are referred to as partial occupied bands (PODs) and partial unoccupied bands (PUDs), respectively. The numbers of partial frequency bands in PODs and PUDs are defined as $N_{\eta_{A}},\;N_{\eta_{B}}$.

In the first chirp-modulated symbol in Figure 3, the estimated chirp pattern is $\phi_{l=2}$; thus, the combined results of power in PODs and PUDs, $\eta_{2,A},\;\eta_{2,B}$, are as follows.
(9)$$\begin{array}{c} \eta_{2,A}=\left(\right|X_{1,2}{|}^{2}+|X_{2,3}{|}^{2}+|X_{3,4}{|}^{2}+|X_{4,1}{|}^{2})/4, \end{array}$$
(10)$$\begin{array}{cc} \eta_{2,B}= & \left(\right|X_{1,1}{|}^{2}+|X_{1,3}{|}^{2}+|X_{1,4}{|}^{2}+|X_{2,1}{|}^{2}+|X_{2,2}{|}^{2}+|X_{2,4}{|}^{2}+|X_{3,1}{|}^{2}+{\left|X_{3,2}\right|}^{2}+\\ & |X_{3,3}{|}^{2}+|X_{4,2}{|}^{2}+|X_{4,3}{|}^{2}+|X_{4,4}{|}^{2})/12. \end{array}$$

#### 4.1.2. Estimation of Power

$\eta_{l,A}$ includes the components of the desired signal, noise, and CCI under the access of the SC system. $\eta_{l,A}$ and $\eta_{l,B}$ are as follows.
(11)$$\begin{array}{c} \eta_{l,A}=\hat{P_{D}}+\hat{P_{n}}+\lambda \hat{P_{I}}+\delta_{A}, \end{array}$$
(12)$$\begin{array}{c} \eta_{l,B}=\hat{P_{n}}+\lambda \hat{P_{I}}+\delta_{B}. \end{array}$$
where $\hat{P_{D}},\;\hat{P_{n}},$ and $\hat{P_{I}}$ are the estimated power of the desired signal, noise, and CCI, respectively. λ∈{0,1} is an indicator that specifies the access conditions for the SC system. When the SC system is accessing and not accessing, λ has 1 and 0, respectively, and $\delta_{A}\;\mathrm{and}\;\delta_{B}$ are residual components. The number of partial spectra is so significant that $\delta_{A}$ and $\delta_{B}$ are negligible.

When $\eta_{l,B}$ is subtracted from $\eta_{l,A}$, the noise and CCI components are reduced. If the residual components $\delta_{A}$ and $\delta_{B}$ are ignored, the power of the desired signal is estimated [45]. To estimate the power of the noise and CCI, The access condition of the SC system should be specified to estimate the power of noise and CCI. We decompose the noise and CCI from $\eta_{l,B}$.

#### 4.1.3. Estimation of Noise Power, CCI Power, and COR

$\eta_{l,B}$ are estimated from the various packets. Subsequently, the probability density function of $\eta_{l,B}$ is obtained by histogram analysis. We assume that the access condition of the SC system is modeled as the independent random access for each packet access of LoRa. Consequently, the probability density function of $\eta_{l,B}$ is modeled by the mixture probability of noise and noise plus CCI. In addition, the noise and CCI are modeled using the Gaussian random process. Based on these assumptions, the probability density function of $\eta_{l,B}$ can be modeled as two Gaussian mixture probability density functions of noise and noise plus CCI [13]. We adopted the expectation–maximization (EM) algorithm [14] to estimate the two probability density functions and mixture rate. The EM algorithm is an iterative scheme suitable for estimating latent variables based on the maximum likelihood estimation method. The mean and deviation of the estimated probability density functions were also estimated, and the means are considered as the estimated average power of the noise, $\hat{P_{n}}$ and noise plus CCI,$\hat{P_{n}}+\hat{P_{I}}$. After subtracting the former from the latter, we estimated the estimated average power of the CCI, $\hat{P_{I}}$. The mixture rate estimated using the EM algorithm was equal to that of the SC system [13]. In addition, the variances of the noise power, $\sigma_{\hat{P_{n}}}^{2}$, and noise plus the CCI, $\sigma_{\hat{P_{I}}+\hat{P_{n}}}^{2}$, were also estimated. After subtracting the former from the latter, we also estimated the variance of the CCI, $\sigma_{\hat{P_{I}}}^{2}$.

In this process, the proposed scheme can estimate the power of the desired signal, CCI, and COR.

### 4.2. Packet Selection

The packet error degrades its accuracy in the signal recovery of the proposed estimation; thus, the packet selection is considered.

Figure 5 illustrates the flow of the considered packet selection, where PEP is the process flow of the proposed estimation illustrated in Figure 4 and the estimated $\eta_{l,A}$ is omitted.

The receiver confirms any error bit in the tentative decoded packet using the cyclic redundancy check (CRC). If the CRC is OK, the tentative decoded packet is adopted to estimate the desired power, CCI power, and COR in the final estimation. If not, the tentative decoded packet is moved to the next step. For simplicity, CRC OK and CRC NG entail that the decoded packet does not include any error bits and includes one or more error bits, respectively. Subsequently, the probability distribution functions of the noise power and CCI plus noise power are estimated by the EM algorithm using only the tentative decoded packets with the CRC OK. After the subtraction, we obtain the average noise power, ${\hat{P_{n}}}^{*}$, average CCI power, ${\hat{P_{I}}}^{*}$, and variance of the CCI power, $\sigma_{\hat{I}}^{2*}$. Note that to distinguish the tentative and conclusive estimations, ∗ is added to each tentatively estimated result.

We subtract ${\hat{P_{n}}}^{*}$ from the combined power in the PUD from the packet with the CRC NG, $\eta_{l,B}$, and obtain the estimated CCI power, $\hat{P_{I}^{\prime }}$. Note that we assume that the reason for the CRC NG is the occurrence of CCI; thus, $\eta_{l,B}$ includes the CCI components. Subsequently, the following equation is confirmed:(13)PI^>PI∗^−σPI^∗
where σPI^∗ denotes the deviation of σPI^2∗. If Equation (Equation 13) is satisfied, the packet from which $\eta_{l,B}$ is estimated is adopted to determine the desired power, CCI power, and COR in the final estimation.

The reason for using Equation (Equation 13) as the selection criterion is as follows: As explained, the dominant factor of the packet error is the CCI. In particular, as the CCI power increases, there is a higher occurrence of the probability of the packet error. If the estimated CCI power exceeds the threshold, which is as large as the average CCI power minus the deviation of the CCI power, we assume that the estimated CCI is most likely true.

Finally, the probability density functions of the noise and CCI plus noise are estimated by the EM algorithm from the packets with the CRC OK and those satisfied by the criteria in Equation (Equation 13). After some subtractive processes, we estimate the noise power, $P_{n}$, CCI power, $P_{I}$, and COR, ρ. In addition, the power of the desired signal power, $\hat{P_{D}}$, is also estimated from the packets with CRC OK and those satisfied by Equation (Equation 13).

## 5. Channel Selection Scheme Based on Estimation of Power and COR

In the channel selection scheme based on the estimation of the signal power, CCI power, and COR, the packet error rate based on the bit error rate [46] is adopted, and is provided as:(14)$$\begin{array}{c} \beta \left(N_{SF},\gamma_{k}\right)=\{1-0.5\times \\ \;\;Q\left(1.28\cdot \sqrt{\Gamma \cdot 2^{N_{SF}}}-1.28\cdot \sqrt{N_{SF}}+0.4\right){\}}^{N}, \end{array}$$
where $N_{pkt}$ is the number of chirp-modulated symbols per packet and $\gamma_{k}$ is the signal to noise plus interference power ratio (SINR) in the *k*th channel. We define $\gamma_{k,I}$ and $\gamma_{k,o}$ as the SINR of the *k*th channel with and without the occurrence of the CCI, respectively. The occurrence of the CCI is equal to the COR of the *k*th channel, $\rho_{k}$. Therefore, the PDR of the *k*th channel is given as
(15)$$\begin{array}{c} U_{k}=\left(1-\rho_{k}\right)\beta \left(N_{SF},\gamma_{k,o}\right)+\rho_{k}\beta \left(N_{SF},\gamma_{k,I}\right). \end{array}$$

The accessing channel is determined by maximizing the PDR; thus, it is constructed as
(16)$$\begin{array}{c} k^{☆}=\arg_{k}\max U_{k} \end{array}$$

Note that the PDR derived by Equation (Equation 15) is decided by the signal power, the interference power, the noise power, and the COR. The estimation errors directly cause the estimation errors of PDR, resulting in the failure of the channel selection with the maximal actual PDR. As a result, the access opportunities of the vacant channel become lower and the packet collision due to the simultaneous access between the sensor of LoRa and the SC system occurs more frequently. The estimation accuracies of the signal power, the interference power, the noise power, and the COR decide the PDR performance.

## 6. Simulation Results

Table 1 presents the simulation parameters of the proposed LoRa system. A wireless channel model is provided by [47].

The received signal power of the receiver accessing the transmitter is provided as
(17)$$\begin{array}{c} P_{rec}=P_{trans}-10a\log_{10}d+b+10c\log_{10}f_{c}+\lambda \end{array}$$
where $P_{trans}$ and $f_{c}$ are the transmit power and center frequency, respectively. The parameters of a,b,andc are presented in Table 1. *d* is the distance between the node and data center and λ is a shadowing component that is modeled by a random variable of the log-normal distribution. In the simulation, we assume that the CRC detection error by CRC does not occur.

This simulation assumes a noise figure in the RF front end is 0 dB. The typical value of the noise figure is around 6 to 8 dB [48]. As the noise figure becomes larger, the coverage, which is the maximal distance between the sensor and the data center for the availability of data gathering, becomes smaller. The evaluation of the impact of the noise figure on coverage reduction will be looked at in future work.

Figure 6 illustrates the probability density function of the noise and CCI in each estimation scheme. Table 2 presents the estimation results for the average power and COR in each estimation scheme. In this evaluation, the interference-to-noise power ratio (INR) is 18.5 dB and the packet error rate is 0.044. In this figure, “All packet use”, “Only CRC OK”, and “Ideal” are the estimation schemes that adopt all the received packets, packets with CRC OK, and all the packets not including any bit errors.

In this figure, the probability density function estimated by the proposed scheme is closest to that of “Ideal”. The reason for the degradation of “All packet use” is the received packets, including the error bits causing missed detections of POD and PUD. In the “Only CRC OK,” the interference is negligible in that the packet error does not occur; thus, the estimated power of the CCI is lesser than the practical one. In the proposed estimation, although packet error occurs, the estimated power of the CCI over a certain threshold is considered to be a highly accurate estimation; thus, the accuracy of the estimation is improved.

Figure 7, Figure 8 and Figure 9 illustrate the performance between the INR and mean percentage error (*MPE*) in the various estimation schemes. The *MPE* is defined as follows:(18)$$\begin{array}{c} MPE=E\left[\frac{\alpha -\hat{\alpha }}{\alpha }\right], \end{array}$$
where E[·] is the expectation operation and $\alpha ,\hat{\alpha }$ are the estimated and true values, respectively.

In these figures, when INR is larger than 12 dB, the performance difference between the proposed estimation and the other estimates is large, and the proposed estimation achieves the best performance. To clarify the performance improvement of the proposed scheme, Figure 10 illustrates the relationship between the INR and packet error rate. Above 12 dB, the packet error rate is greater than 0.1. The impact of the packet error on the estimation increases. In the proposed scheme, even if a packet error occurs, the estimated power of the CCI over a certain threshold is considered highly accurate; therefore, the impact of the packet error can be mitigated. When the INR is larger than 18 dB, the packet error rate becomes 1.0. If CRC confirms all the packets contain errors during the tentative demodulation, the proposed estimation scheme cannot work. The operation region of the proposed estimation scheme is limited to INR 18dB or smaller.

We evaluated the performance of channel selection based on each estimation scheme. The simulation parameters are the same as those in Table 1, and their differences are as follows: There are five channels and one SC system accesses each channel; thus, there are five SC systems. The distance between the transmitter of the SC system and data center is a random variable drawn from an independent uniform distribution between 500 and 900 m. Each SC system is an independent random access, where the access probability to the channel, which is equal to the COR of the channel, is modeled by a uniform distribution from 0 to 1.0.

Figure 11 illustrates the cumulative distribution function (CDF) of PDR for various channel selection schemes. From this figure, the proposed scheme is better than other conventional schemes, and it is seven points worse than the “Ideal”. In CDF = 0.1, the PDR of the proposed scheme is 10 and 20 points better than those of “All packet use” and “Only CRC OK”, where the point is defined as the difference of two percentage values. The reason for the better PDR achieved by the proposed scheme is the accurate estimation of signal, noise, and interference power and the COR. From Figure 7, Figure 8 and Figure 9, the proposed estimation achieves a better performance in terms of the estimation accuracy than “All packet use” and “Only CRC OK”. As we explain in Section 5, the estimation error of power and COR cause the estimation error of the PDR. As a result, “All packet use” and “Only CRC OK” select the channel with the lower actual PDR than that with the largest one because of the estimation error of PDR. From this result, we confirm that the channel selection with the proposed scheme achieves a high PDR performance.

## 7. Conclusions

This study proposes an estimation scheme for the desired signal power, co-channel interference (CCI) power, and channel occupancy ratio (COR) in a long-range (LoRa) wireless communication system. In the proposed scheme, the chirp modulation scheme employed in LoRa occupied the partial frequency band, and the proposed scheme estimated the desired signal power and noise plus the CCI power or the noise power separately from each partial frequency band. To distinguish between the noise and noise in addition to the CCI power, the expectation maximization algorithm estimated the probability density functions of the noise and noise plus CCI. In addition, it estimated the mixture rate of the two functions; thus, it estimated the COR which was equal to the estimated mixture rate. To mitigate the impact of the packet error, the criterion of estimation accuracy using the estimated CCI power was considered. If the criterion is satisfied, the received packet is utilized to estimate the power and COR even if a packet error occurs. From the computer simulation, we clarified the advantages of the proposed scheme in terms of not only the estimation accuracy of the power and COR but also the channel selection for improving the packet delivery rate.

This paper assumes the access of each sensor is the time-division manner. If more sensors are deployed, the number of sensors per channel becomes so large. Then, the independent time interval cannot be assigned to each sensor even under the 1% duty cycle [44]. To accommodate the access from many sensors, the channel selection is not only based on the COR of the other system, but also the access traffic from the sensors of the own system is necessary. The introduction of such issues to the proposed estimation scheme will be a part of important future work.

## Figures and Tables

**Figure 1 sensors-22-04140-f001:**
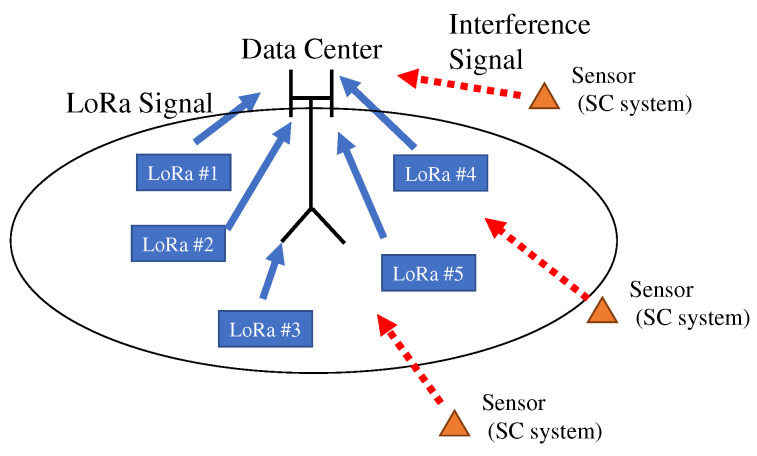
System overview of assumed wireless sensor networks. Frequency spectrum sharing between LoRa and single-carrier systems was considered.

**Figure 2 sensors-22-04140-f002:**
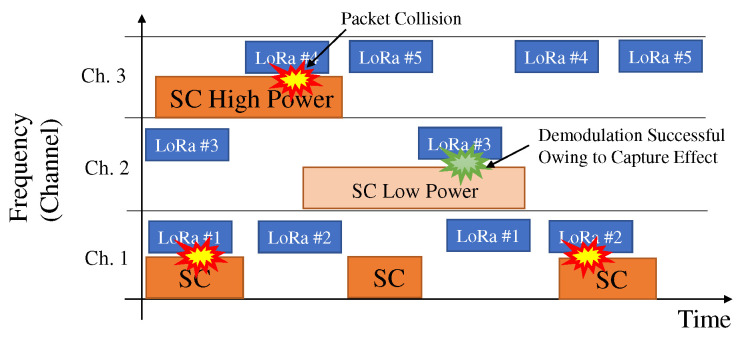
Accessing image of the multichannel environment under spectrum sharing between LoRa and the SC system.

**Figure 3 sensors-22-04140-f003:**
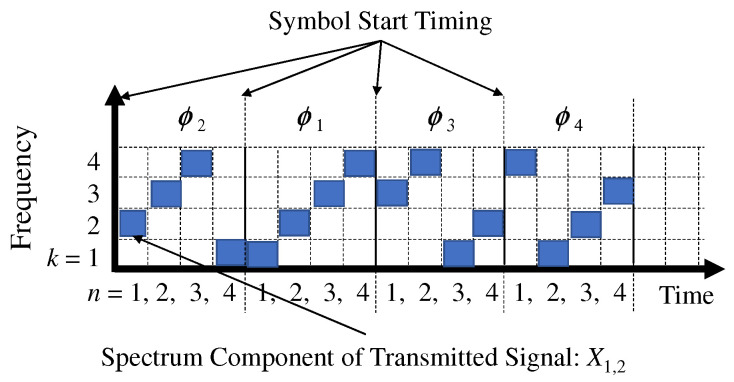
Frequency spectrum of chirp modulation in LoRa. Carrier frequency cyclically increases or decreases.

**Figure 4 sensors-22-04140-f004:**
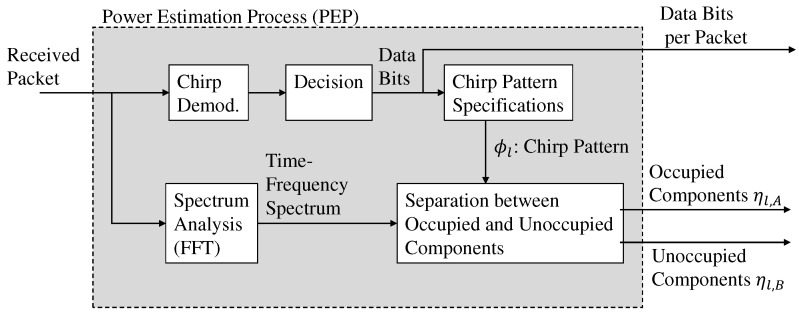
System flow of the power estimation process (PEP) in the proposed estimation scheme.

**Figure 5 sensors-22-04140-f005:**
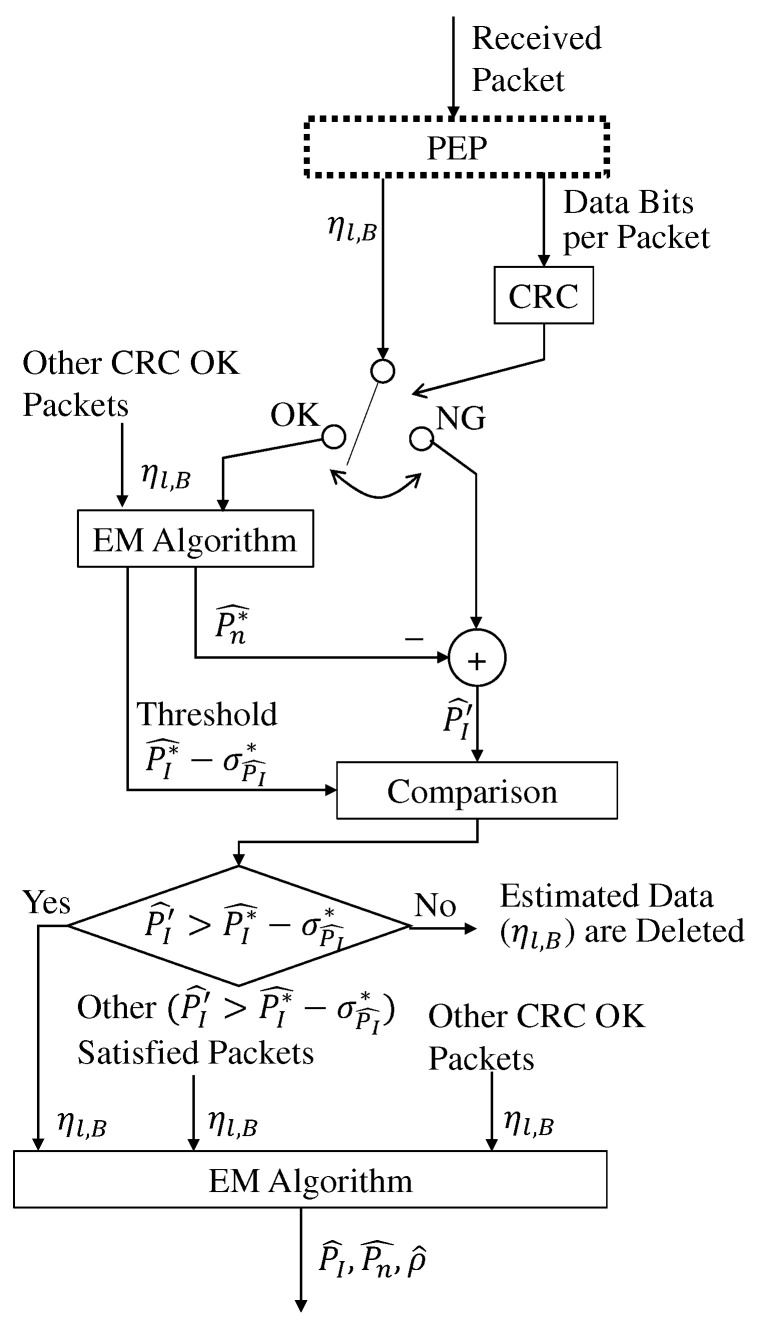
Flow for packet selection in proposed estimation scheme.

**Figure 6 sensors-22-04140-f006:**
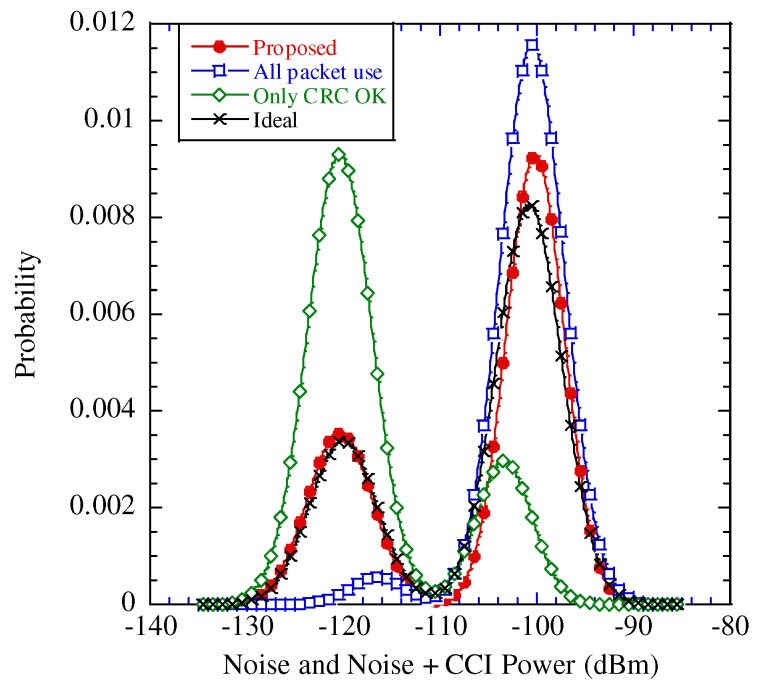
Example of estimating noise and CCI plus the noise power distribution in each estimation scheme and ideal.

**Figure 7 sensors-22-04140-f007:**
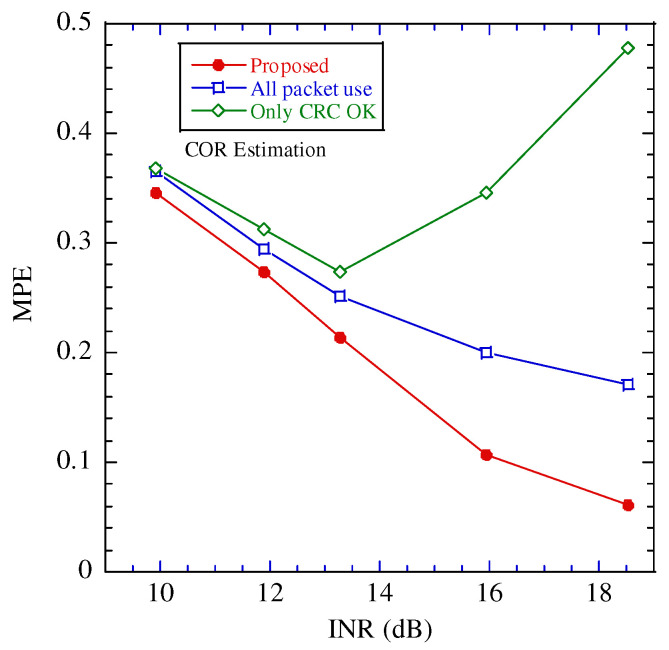
Performance between MPE of estimating the channel occupancy rate (COR) and interference to noise power ratio (INR).

**Figure 8 sensors-22-04140-f008:**
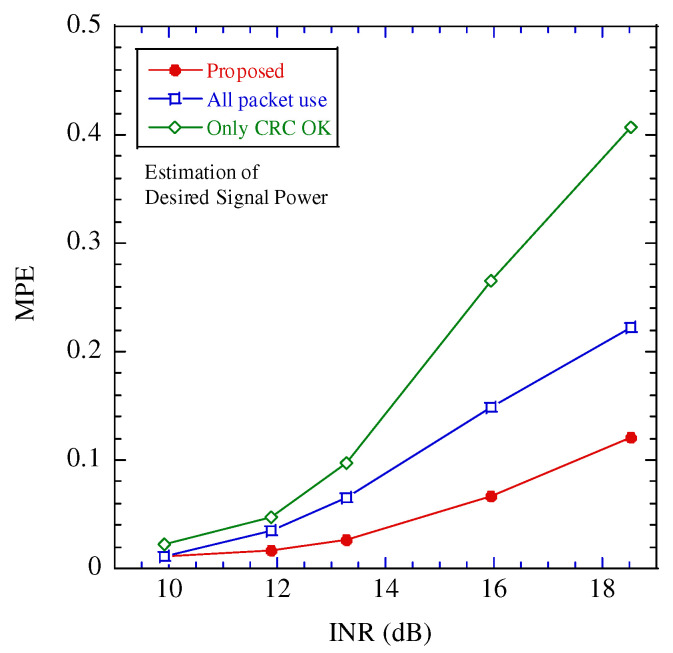
Performance between MPE of estimating the desired signal power and interference to noise power ratio (INR).

**Figure 9 sensors-22-04140-f009:**
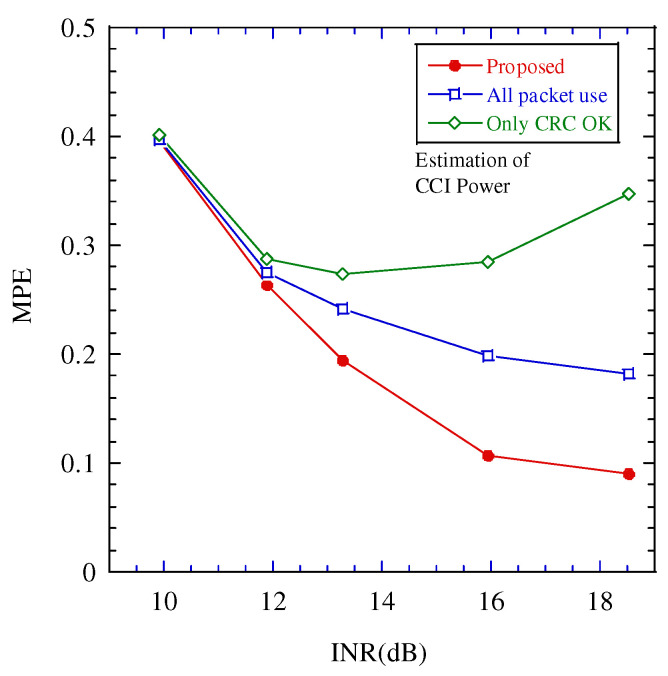
Performance between MPE of estimating the CCI power and interference to noise power ratio (INR).

**Figure 10 sensors-22-04140-f010:**
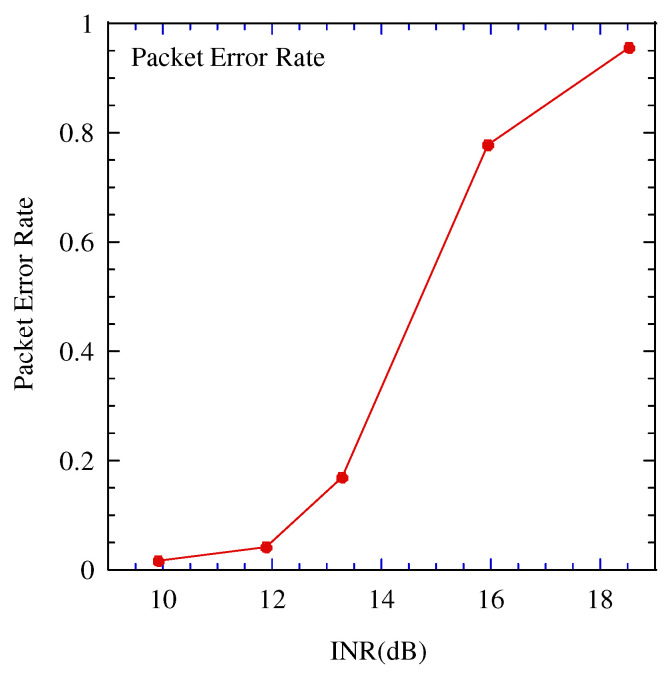
Performance between the packet error rate in the tentative demodulation of LoRa and the interference to noise power ratio (INR).

**Figure 11 sensors-22-04140-f011:**
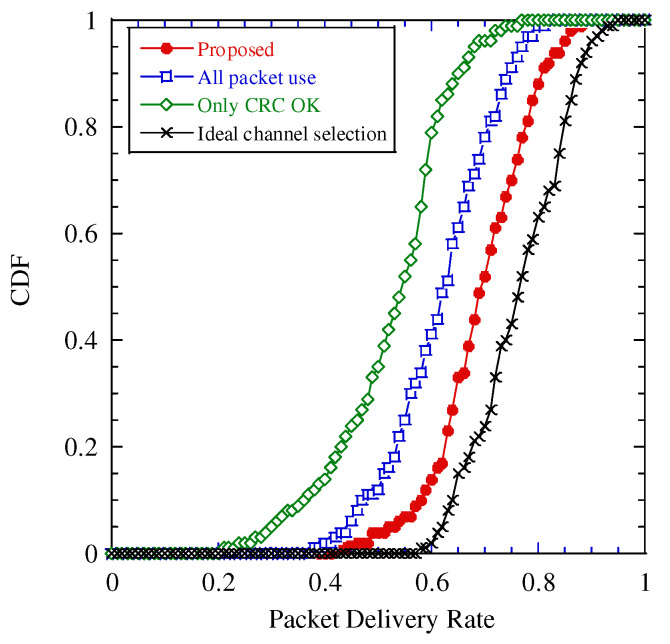
Cumulative distribution function (CDF) of the packet delivery rate.

**Table 1 sensors-22-04140-t001:** Wireless system parameter in computer simulation.

Center frequency $f_{c}$	923 MHz
Spreading factor, $N_{SF}$	7
LoRa bandwidth	250 kHz
SC Bandwidth	200 kHz
LoRa and SC transmission power $P_{trans}$	13 dBm
Number of packets for estimation	1000
Number of symbols per packet, $N_{pkt}$	200
COR of SC	0.7
Distance between data centers and sensor in LoRa	500 m
Distance between data centers of LoRa and SC transmitter	500 m to 900 m
Propagation coefficient of the distance loss, *a* [47]	4
Propagation offset, *b* [47]	9.5
Propagation coefficient of the frequency loss, *c* [47]	4.5
Transmitted antenna gain	0 dBi
Received antenna gain	16 dBi
Shadowing deviation [47]	3.48 dB
Shadowing coefficient	0
Noise power-spectrum density	−174 dBm/Hz
Amplification rate of the low-noise amplifier	60 dB
Noise figure	0 dB

**Table 2 sensors-22-04140-t002:** Results of the estimated power and COR in each estimation scheme.

Estimating Terms	Ideal	Proposed	All Packets Use	Only CRC OK
Noise + CCI power	−101 dBm	−100 dBm	−100 dBm	−103 dBm
Noise	−120 dBm	−120 dBm	−117 dBm	−120 dBm
COR	0.70	0.70	0.96	0.22

## Data Availability

Not applicable.

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
