# Peer review of "Estimation Based on Chirp Modulation for Desired and Interference Power and Channel Occupancy Ratio in LoRa"

_sensors, 2022, doi:10.3390/s22114140_

Round 1
Reviewer 1 Report
This paper proposes a desired signal power, co-channel interference and channel occupancy ratio estimation method for LoRa modulation. The proposed method is based on the estimation of chirp pattern according to the transmitted information from the tentative chirp demodulation. partially occupied bands (POD) and partially unoccupied bands (PUD) are firstly estimated from received signal spectrum. Further, desired signal power, co-channel interference and channel occupancy ratio are estimated by using estimated POD and PUD to select an appropriate channel to reduce the packet error. The authors evaluated performance of the proposed concept by computer simulation.
The organization of this paper seems well, however, the reviewer has a couple of concerns to publish paper as it is.
The following mandatory changes should be considered:
(1) (Table 1)
(1-1)"Propagation coefficient of the distance loss" should be 'a' (it is missing).
(1-2) Though amplification rate of the low-noise amplifier is listed in the table, noise figure of low-noise amplifier is not taken into account in the simulation. This noise figure may make an impact to overall estimation performance because it makes additional noise level at amplifier output.
(2) Consider to include discussions related to Figure 10 into discussions on Figure 11. Since Figure 10 seems just an example of packet delivery rate, it looks difficult to find the specific meanings to show this plot for discussion.
(3) Meaning of "points" should be clarified. (e.g., line 355 (p.12), line 361 (p.,13))
Author Response
--------------------------
Reviewer 1
--------------------------
Comments and Suggestions for Authors
This paper proposes a desired signal power, co-channel interference and channel occupancy ratio estimation method for LoRa modulation. The proposed method is based on the estimation of chirp pattern according to the transmitted information from the tentative chirp demodulation. partially occupied bands (POD) and partially unoccupied bands (PUD) are firstly estimated from received signal spectrum. Further, desired signal power, co-channel interference and channel occupancy ratio are estimated by using estimated POD and PUD to select an appropriate channel to reduce the packet error. The authors evaluated performance of the proposed concept by computer simulation.
The organization of this paper seems well, however, the reviewer has a couple of concerns to publish paper as it is. The following mandatory changes should be considered:
(Reply)
We greatly appreciate spending your precious time to review our paper and positive comment. We have revised the paper to take into account the comment.
(R1-1) (Table 1)
(R1-1-1)"Propagation coefficient of the distance loss" should be 'a' (it is missing).
(R1-1-2) Though amplification rate of the low-noise amplifier is listed in the table, noise figure of low-noise amplifier is not taken into account in the simulation. This noise figure may make an impact to overall estimation performance because it makes additional noise level at amplifier output.
(Reply)
"Propagation coefficient of the distance loss", $a$, is added to Table 1.
As the noise figure of the RF front end becomes larger, the coverage area, which is the maximal distance between the sensor and the data center for the availability of data gathering, becomes smaller. The typical value of the noise figure is around 6 dB to 8 dB [razavi], which is significantly smaller than a propagation distance loss. Therefore, the impact of the noise figure on coverage reduction is not so significant. In the computer simulation, we consider the access from the sensor within the coverage to the data center. Therefore, the noise figure value does not dramatically change the performance of estimating power and channel occupancy ratio. Thus, there is no direct impact on overall estimation performance.
The revised paper explicitly describes the noise figure in the RF front end as 0 dB. The evaluation of the impact of the noise figure on coverage reduction is left as a future work.
Thank you very much for your valuable comments.
[razavi]B. Razavi, RF Microelectronics, 2nd edition, Prentice Hall, 2012
(R1-2) Consider to include discussions related to Figure 10 into discussions on Figure 11. Since Figure 10 seems just an example of packet delivery rate, it looks difficult to find the specific meanings to show this plot for discussion.
(Reply)
Thank you for pointing out this issue. As the reviewer pointed out, Fig. 10 shows a snapshot of the packet delivery rate performance. We have removed Fig. 10 and the corresponding explanations in the revised manuscript. Instead, we have added a discussion about the qualitative reason why the proposed estimation method is better than the conventional methods by referring to Fig. 11.
Thank you very much for your valuable comments.
(R1-3) Meaning of "points" should be clarified. (e.g., line 355 (p.12), line 361 (p.,13))
(Reply)
We have added the definition of “points” in the revised paper.
Reviewer 2 Report
In this paper, the authors proposed an estimation scheme for the desired signal power, co-channel interference (CCI) power, and channel occupancy ratio (COR) in a long-range (LoRa) wireless communication system. The novelty in the paper is poor, but the presentation and organization of the manuscript is good. However, the following corrections to be made before consider this paper for publication.
- The citations provided for the list of contributions in the Introductions seems the paper is acquire the knowledge from the citations. In that case, what is the novelty in the paper.
- The literature of the paper is poor. The authors must provide the literature on recently published papers.
- The system model is not clearly described. The system model section can be elaborated.
- The reasons for achieving the superior in terms of performance of the existing systems to be mentioned in the paper. It is also recommended to provide the limitations.
- How this work can be extended in the near future?
- The captions of the figures are elaborated. The authors mentioned a short labels, recommended to provide a detailed labels for them.
Author Response
--------------------------
Reviewer 2
--------------------------
In this paper, the authors proposed an estimation scheme for the desired signal power, co-channel interference (CCI) power, and channel occupancy ratio (COR) in a long-range (LoRa) wireless communication system. The novelty in the paper is poor, but the presentation and organization of the manuscript is good. However, the following corrections to be made before consider this paper for publication.
(Reply)
We greatly appreciate spending your precious time to review our paper and positive comment. We have revised the paper to take into account the comment.
(R2-1) The citations provided for the list of contributions in the Introductions seems the paper is acquire the knowledge from the citations. In that case, what is the novelty in the paper.
The literature of the paper is poor. The authors must provide the literature on recently published papers.
(Reply)
We are sorry for the insufficient explanation of the contributions of this manuscript in the original version. In the revised manuscript, we clarify the novelty and contributions in the introduction as follows.
- An estimation scheme of the desired signal power, CCI power, and COR based on the chirp modulation of LoRa is proposed.
- A packet selection based on the estimated CCI power is proposed to improve the estimation accuracy.
- A channel selection criterion for maximizing the PDR performance is constructed.
Although we have proposed the estimation scheme in our prior works [kobayashi1][kobayashi2][kobayashi3], we provide additional explanations to understand the principle of the proposed scheme and extensive simulation performance evaluations. In the revised manuscript, we clearly explain the contributions of the manuscript.
Furthermore, to solidify the contributions of the manuscript, we have added related works in the past three years in Section 2.
Thank you very much for your valuable comments.
[kobayashi1]G. Kobayashi, O. Takyu, K. Adachi, M. Ohta and T. Fujii, "Estimation of Desired power and Undesired power Using Chirp Demodulation and Evaluation of Accuracy," 2020 Asia-Pacific Signal and Information Processing Association Annual Summit and Conference (APSIPA ASC), Auckland, New Zealand, 2020, pp. 1513-1518.
[kobayashi2]G. Kobayashi, O. Takyu, K. Adachi, M. Ohta and T. Fujii, "Proposal of interference power occupancy estimation method using chirp demodulation," 2021 Twelfth International Conference on Ubiquitous and Future Networks (ICUFN), Jeju Island, Korea, Republic of, 2021, pp. 187-191.
[kobayashi3]Gaku Kobayashi, Osamu Takyu, Koichi Adachi, Mai Ohta, Takeo Fujii "Estimation Accuracy to Channel Occupancy Ratio and Signal Power under Co-Channel Interference in LoRa" IEICE ICETC 2021, Dec. 2021.
(R2-2) The system model is not clearly described. The system model section can be elaborated.
(Reply)
We have modified the figure of the system model for better understanding of the assumed LoRa system and the frequency spectrum sharing between the LoRa and the single carrier (SC) system. In addition, the image of wireless access scheme among sensors and the SC system is added to the revised paper.
Thank you very much for your valuable comments.
(R2-3) The reasons for achieving the superior in terms of performance of the existing systems to be mentioned in the paper. It is also recommended to provide the limitations.
(Reply)
The channel selection scheme based on the maximal packet delivery rate (PDR) criterion needs to calculate the theoretical PDR defined by equations (14) to (16). The channel occupancy ratio (COR) of the SC system and the signal to noise plus interference power ratio ($\Gamma$) should be estimated for deriving PDR. Therefore, the estimation error of the COR, desired signal power, interference power, and noise power cause the estimation error of PDR, which results in the failure of the channel selection with the maximal actual PDR. As a result, the access opportunities of the vacant channel become lower and the packet collision due to the simultaneous access between the sensor of LoRa and the SC system occurs more frequently. Therefore, the PDR performance is degraded.
In the revised manuscript, we have added the reasons why the proposed estimation scheme can achieve better PDR performance than the conventional schemes by the relationship between the calculated PDR and the estimated COR and powers.
As the reviewer suggested, we have revised the manuscript to show the limitations of the proposed scheme. If CRC confirms all the packets contains error during the tentative demodulation, the proposed estimation scheme cannot work. Figure 9 shows that the packet error rate (PER) performance becomes 1.0 when the interference to noise power ratio (INR) is over 18 dB. Thus, the operation region of the proposed estimation scheme is limited to INR 18dB or smaller.
Thank you very much for your valuable comments.
(R2-4) How this work can be extended in the near future?
(Reply)
This paper assumes the access of each sensor within a channel is the time division manner. If more sensors are deployed, the number of sensors per channel becomes so large. Then, the independent time interval cannot be assigned to each sensor even under the 1$\%$ duty cycle $\%$ [LoRa_Specifies]. To accommodate the access from many sensors, the channel selection based on not only the COR of the other system but also the access traffic from the sensors of the own system is necessary. The introduction of such issues to the proposed estimation scheme is an important future work. We have added this issue in Conclusion.
[LoRa_Specifies]RP2-1.0.1 LoRaWAN® Regional Parameters,
https://lora-alliance.org/resource_hub/rp2-101-lorawan-regional-parameters-2/
(R2-5) The captions of the figures are elaborated. The authors mentioned a short labels, recommended to provide a detailed labels for them.
(Reply)
The detailed explanation of the caption in each figure is added in the revised paper.
Reviewer 3 Report
Please see the attached file.

Author Response
--------------------------
Reviewer 3
--------------------------
This study presented an estimation system for the desired signal power, co-channel interference (CCI) power, and channel occupancy ratio (COR) in a long-range (LoRa) wireless communication system. This work presents a very important added value for research in this field of wireless communication.
The proposed method in the system used the chirp modulations in LoRa occupied the partial frequency band, and the proposed scheme estimated the desired signal power and noise plus the CCI power or the noise power separately from beach partial frequency band. To distinguish between the noise and noise in addition to the CCI power, the expectation maximization algorithm estimated the probability density functions of the noise and noise plus CCI.
In addition, it estimated the mixture rate of the two functions; thus, it estimated the COR which was equal to the estimated mixture rate. To mitigate the impact of packet error, the criterion of estimation accuracy using the estimated CCI power was considered. If the criterion is satisfied, the received packet is utilized to estimate the power and COR even if a packet error occurs.
(Reply)
We greatly appreciate spending your precious time to review our paper and positive comment. We have revised the paper to take into account the comment.
(R3-1) The paper deals with a very important topic that is the spectrum efficiency and management, but the work does not situate the solution in the presence of 5g and 6g technologies.
(Reply)
The wireless sensor networks constructed by the cellular system are also attracting attention, such as massive machine-type communications for 5G [mMTC] and narrow-band IoT [NB_IoT]. However, their high running cost and the requirement of a wireless license are the main hindrances to their deployment. On the other hand, the LoRa has an advantage in terms of easy deployment due to the use of unlicensed bands and low cost.
The revised paper shows the difference between the LPWAN and the wireless sensor networks constructed by the cellular systems.
Thank you very much for your valuable comments.
[mMTC] C. Bockelmann et al., "Towards Massive Connectivity Support for Scalable mMTC Communications in 5G Networks," in IEEE Access, vol. 6, pp. 28969-28992, 2018.
[NB_IoT] E. M. Migabo, K. D. Djouani and A. M. Kurien, "The Narrowband Internet of Things (NB-IoT) Resources Management Performance State of Art, Challenges, and Opportunities," in IEEE Access, vol. 8, pp. 97658-97675, 2020.
(R3-2) The liste of references must be enhanced by introducing a new works in the field.
(Reply)
The related works of the past three years are added to Section 2 in the revised paper for clarifying the difference between the conventional works and our work.
Thank you very much for your valuable comments.
Round 2
Reviewer 1 Report
Overall quality of the manuscript has been improved very much in the revised version. The reviewer agrees to publish this paper as it is.
Reviewer 2 Report
The authors addressed all the recommended comments. The current version is well improved. So, I can recommend this version of the manuscript for the publication in this journal. Congratulations to the authors.